# *Millepora* spp. as Substrates of Their Hydrozoan Counterparts *Stylaster* sp. in the Pacific Ocean

**Chloé Julie Loïs Fourreau** [1,*] , **Daniela Pica** [2] , **Emmeline A. Jamodiong** [1] , **Guillermo Mironenko Castelló** [1] , **Iori Mizukami** [1] **and James Davis Reimer** [1,3]

1 Molecular Invertebrate Systematics and Ecology Laboratory, Graduate School of Engineering and Science, University of the Ryukyus, 1 Senbaru, Nishihara 903-0123, Okinawa, Japan; emmeline.4993@yahoo.com (E.A.J.); gui.mironenko@gmail.com (G.M.C.); iori.mizukami@my.jcu.edu.au (I.M.); jreimer@sci.u-ryukyu.ac.jp (J.D.R.)

2 Department of Integrative Marine Ecology, Stazione Zoologica Anton Dohrn, Calabria Marine Centre, 87071 Amendolara, Italy; daniela.pica@szn.it

3 Tropical Biosphere Research Center, University of the Ryukyus, 1 Senbaru, Nishihara 903-0123, Okinawa, Japan

* Correspondence: chloisf@gmail.com

**Abstract:** The association between two hydrozoans, *Stylaster* sp. and *Millepora* spp., has been described as a case of pseudo-auto-epizoism, and has only been reported from the Caribbean region of the Atlantic Ocean. Here, we report on the occurrence of this association in the Pacific Ocean on coral reefs around Iriomote-jima Island, Japan, suggesting the association to be more widespread than had previously been thought. Moreover, *Stylaster* sp. colonies were observed living healthily on bleached and dead branches of *Millepora* spp., indicating that this interaction is facultative. The interaction reported here differs from the relationship between the Caribbean *Stylaster roseus* and *Millepora alcicornis* by the connection points between the two partners, which is made evident by the whitening of the *Millepora* counterpart in Iriomote-jima Island, while being seamless in the Caribbean association. Further research is necessary to fully understand the nature of these relationships, comprehending under what conditions it occurs, and establishing which species are involved in the interactions.

**Keywords:** Hydrozoa; Milleporidae; Stylasteridae; coral reefs; epizoism; species association





## 1. Introduction

Species interactions play an important role in shaping coral reef ecosystems and diversity. Skeleton-forming (scleractinian) corals are known to be associated with a plethora of invertebrate fauna [1] and their three-dimensional structure provides shelter to other organisms. Similar to scleractinians, hydrozoan corals host a wide range of taxa, and in particular, fire corals of the genus *Millepora* host members of many marine phyla among their associated fauna [2], including cnidarians, crustaceans, molluscs, polychaetes, tunicates, sponges, and fish [3]. One notable epibiont of *Millepora* is another hydrozoan species, *Stylaster roseus* (Pallas, 1766), which has been found to live attached to *Millepora alcicornis* Linnaeus, 1758 in the Caribbean [4,5], despite it usually inhabiting small crevices and dim environments. This apparently rare interaction has been more extensively recorded in the coral reefs of Bonaire [6]. The relationship was described as a rare example of pseudo-auto-epizoism, a relationship in which a species lives on the surface of a closely related host, using it as a substratum [7]. Both organisms belong to groups of hydrozoans that calcify, namely the families Stylasteridae Gray, 1847 and Milleporidae Fleming, 1828 [7]. In the biodiverse marine ecosystems of the Ryukyu Archipelago, several unexpected interactions between different taxa have been reported [8–10], and some of these associations have been found to be widespread despite having remained unreported for a substantial amount of

time [11]. Here we report our observations of the association between *Millepora* spp. and *Stylaster* sp. in the subtropical island of Iriomote-jima, Okinawa, Japan, constituting the first record of such an interaction between *Millepora* and *Stylaster* in the Pacific Ocean.

## 2. Materials and Methods

Three sites on the north of Iriomote-jima Island, Okinawa, Japan were visited by SCUBA diving where the association of *Stylaster* and *Millepora* was observed and photographed with an Olympus TG6 camera: at 5 to 7 m depths on a reef in front of Funauki Port lighthouse (24.35184° N, 123.70311° E) in September 2022, at approximately 5 m depth at Nishizaki-Higashi point (24.43884° N, 123.78771° E) in March 2023, and at 8 m depth in Nakano Beach (24.43151° N, 123.79237° E) in June 2023. Additionally, benthic transects (3 × 25 m/site) were conducted at four sites around Iriomote-jiima Island (Nakano Beach as mentioned above, and three additional sites: Nakano-oki (24.43522° N, 123.79934° E), Amitori (24.34577° N, 123.6899° E), and Sotopanari (24.38227° N, 123.71964° E), all depths 5 to 10 m). To evaluate the prevalence of the association, the number of *Millepora* spp. colonies were counted on each transect, and for each of them the presence or absence of *Stylaster* sp. was noted. Using R ver. 1.4 and package stats [12], we performed a one-way ANOVA to determine whether there were significant differences in the abundance of *Millepora* between sites, and to determine whether there were significant differences in abundance of *Millepora* hosting *Stylaster* between sites.

## 3. Results and Discussion

The association between *Stylaster* sp. and *Millepora* spp. was observed in three sites around Iriomote-jima Island: Funauki Port Lighthouse, Nishizaki-Higashi and Nakano Beach (Figure 1).

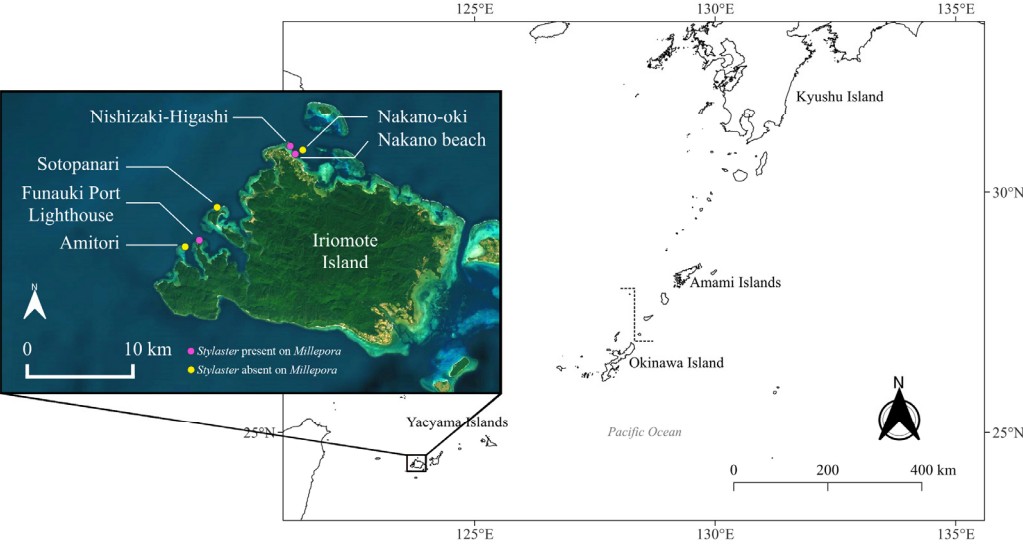

**Figure 1.** Map of sites in Iriomote-jima Island, Japan, with sites where *Stylaster* sp. Was detected on *Millepora* spp. in pink, and sites where *Stylaster* sp. Was not observed on *Millepora* spp. in yellow.

Although samples of *Millepora* spp. and *Stylaster* sp. could not be collected, the observation of the association in situ provided some insights into the relationship between the two partners and allowed for a comparison with the association between *Millepora* and *Stylaster* that has been reported in the Caribbean. In Iriomote-jima, the observed colonies of *Stylaster* sp. were growing downward from the underside of branches of *Millepora* (Figures 2a and 3a). Numerous healthy *Stylaster* sp. colonies could be found on the same *Millepora* colonies (Figure 2a), and this interaction was present on multiple *Millepora* colonies at the three sites where the association was noticed. In addition, colonies of *Stylaster* sp. were observed on bleached *Millepora* (Figure 3) during an intense bleaching

event in September 2022. *Stylaster* sp. colonies were seemingly unaffected by the conditions of their host, as live, open polyps could be observed (Figure 3b). Subsequently, colonies of *Stylaster* sp. were also observed in March 2023 (although at a different site), growing on dead skeletonized branches of *Millepora* (Figure 4a,b). Parts of these *Millepora* colonies showed growth of other benthic components such as crustose coralline algae, ascidians, and polychaete tubes (Figure 4a,b), suggesting that those branches may have been dead for some time. Therefore, it appears that the colonies of *Stylaster* sp. can maintain themselves on unhealthy or long dead *Millepora* colonies. This reinforces the idea that *Stylaster* sp. mainly uses *Millepora* as a substrate, that the species involved may also be found on other substrates, and that the association is facultative, much like *Stylaster roseus* in the Caribbean [6]. By using this additional niche, *Stylaster* sp. may be able to reduce intraspecific competition for crevices [6]. However, this does not discount the possibility that the association may have other benefits for *Stylaster* sp., such as an advantage in terms of planktivory by benefiting from the fire coral's position in the water column, as was previously hypothesized by Montano et al. [6]. A notable difference between the association reported herein from Iriomote-jima and the association reported from the Caribbean resides in the appearance of the connecting points between colonies of the two hydrozoan partners. In Bonaire, *S. roseus* was observed in intimately close contact with *M. alcicornis* at the base of the colony, with almost no visible boundary between their tissues [6]. This was interpreted as a sign that the presence of *S. roseus* on *M. alcicornis* did not cause stress to the host. In contrast, the tissue of *Millepora* spp. surrounding *Stylaster* sp. colonies observed in Iriomote-jima appeared to be whitened, and sometimes other benthic components such as calcifying algae, filamentous algae, ascidians, and polychaete tubes could be observed growing on branches of *Millepora* in the vicinity of *Stylaster* sp. (Figures 2a and 4c,d). It is possible that those *Stylaster* sp. colonies have grown from points where branches of *Millepora* had been broken. Another possibility is that *Millepora* spp., while overgrowing dead coral skeletons, may have encountered previously settled *Stylaster* sp. and surrounded these colonies without coming into direct contact with them. Such scenarios may be indicated by parts of the *Millepora* colony seemingly growing on dead corals, highlighted by red circles in Figure 2. Alternatively, direct settlement of *Stylaster* sp. on *Millepora* spp. may be possible and accompanied by a harmful effect on the *Millepora* tissue. In any case, the small distance observed between tissues of the two partners suggests that *Stylaster* sp. is not as well tolerated by *Millepora* in Iriomote-jima as in the case of the Caribbean species duet. In a remarkable case of behavioural adaptation, it has been shown that cyprid larvae of *Wanella milleporae*—a species of barnacle living on *Millepora*—are able to settle directly on the fire coral and inactivate or withstand the corals' nematocysts [13], showing that symbionts develop complex strategies in order to settle on *Millepora* hosts. As there seem to be differences between the *Stylaster*–*Millepora* interactions from Iriomote-jima and those from the Caribbean, these pairs of species represent interesting models to study the interactions and settlement of symbiotic organisms. Further investigations are required to understand the circumstances of the settlement of *Stylaster* sp. on *Millepora* spp.

Although Stylasteridae is considered to be especially diverse in the Central Indo-Pacific region [7], very scant information is available about shallow water Indo-Pacific stylasterids, particularly those from Japan. Without detailed morphological analyses, the identification of the *Stylaster* species was not possible. On the other hand, five species of *Millepora* have been reported in the Archipelago of the Ryukyus and their genetic diversity has been studied, together with aspects of their colony morphology [14]. The *Millepora* colonies that we observed harbouring *Stylaster* had a morphology that was most similar to that of *Millepora intricata* Edwards, 1857 (Figures 2–4), with spaced branches growing in all directions, although the branches appeared thicker than in a recent description of the species [15]. However, in molecular diversity analyses of Ryukyuan *Millepora* spp. made by Takama et al. [14], one clade (Clade 1) harboured most of the branching specimens, lumping the three *Millepora* species with branching growth forms present in the Ryukyus into one group (*M. intricata, M. dichotoma* Forskål, 1775 and *M. tenera* Boschma, 1949), suggesting the

taxonomy of *Millepora* present in the Ryukyus requires further consideration. Based on this information and our data being limited to photographs, we cannot establish the identity of the *Millepora* species we observed associating with *Stylaster* species with certainty. Although we could not identify *Millepora* to species level, it is clear that the set of species involved here are different than in the previously reported relationship, as both *M. alcicornis* and *S. roseus* are Caribbean species [16,17]. Species with similar habits of symbiosis with *Millepora* have been found to exhibit profound genetic differentiation across different regions. Such cases have been documented in several different barnacles associated with *Millepora* [18,19], for which the species are also closely related and morphologically similar.

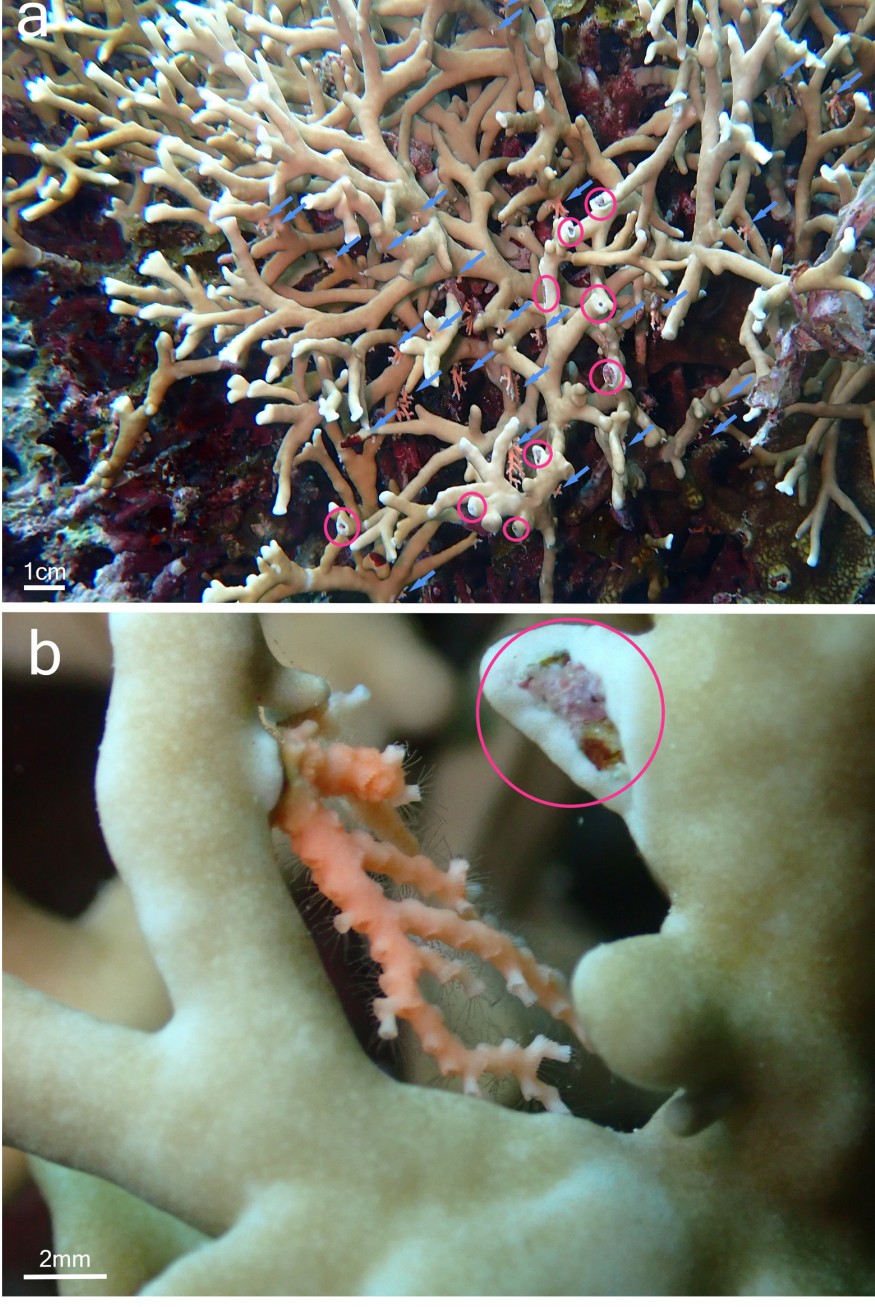

**Figure 2.** *Stylaster* sp. on *Millepora* sp. colony, observed at Nishizaki-Higashi, Iriomote-jima, Japan. (**a**) View of the colony, with blue arrows pointing to *Stylaster* sp. colonies and red circles showing the regions where *Millepora* sp. tissue surrounds benthic component; (**b**) close-up view of a *Stylaster* sp. colony with open polyps close to *Millepora* sp. tissue.

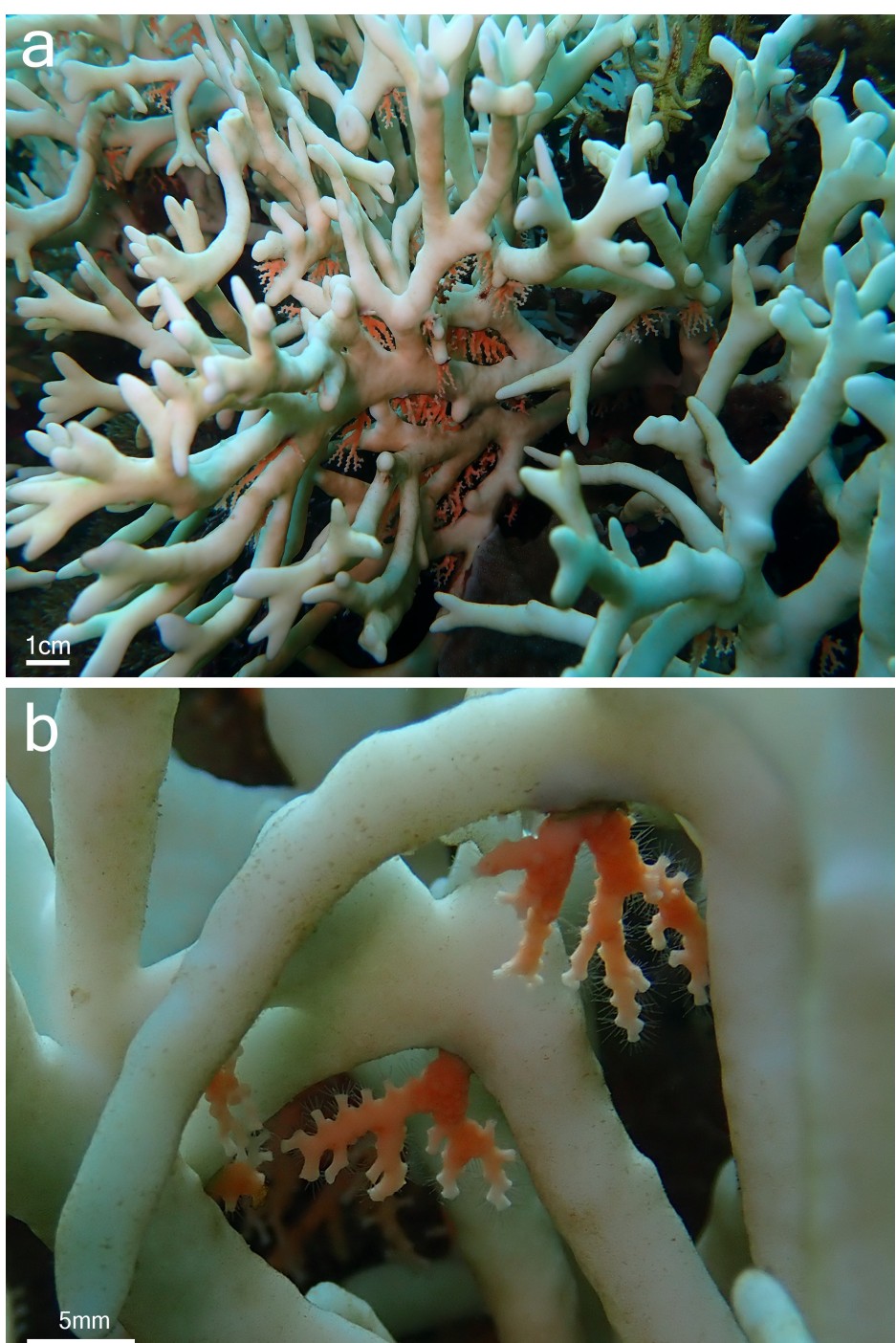

**Figure 3.** *Stylaster* sp. on bleached *Millepora* sp. colony, observed at Funauki Port Lighthouse, Iriomote-jima, Japan. (**a**) View of the colony; (**b**) close-up view at *Stylaster* sp. colonies with open polyps close to *Millepora* sp. tissue.

The fact that the association between *Stylaster* and *Millepora* occurs between different species across two oceans suggests that the physiological and/or genetic background allowing them to associate with one another has been present for a long period of time, or that, alternatively, this type of relationship may have evolved separately in both the Atlantic and Pacific Oceans as a case of parallel evolution. Either way, as the relationship between *Millepora* and *Stylaster* has been defined as a case of pseudo-auto-epizoism, the emergence of such an interaction is likely facilitated by the relatively close phylogenetic

relatedness of the two members [20]. The use of *Millepora* as a substratum for *Stylaster* highlights the adaptation capabilities of stylasterids. Stylasteridae is especially known to be diversified in the deep sea [21] and thought to have conquered shallow-water environments multiple times [21]. The *Millepora–Stylaster* association may be one of the strategies that has helped stylasterids to maintain themselves in highly competitive coral reef environments. Finally, it is notable that, according to a past reconstruction of the group's distribution [22], members of the family Stylasteridae appeared to be primarily distributed in oceanic islands, archipelagos, atolls, or seamounts, and were found to not be as prevalent in waters surrounding large land masses. In the Pacific Ocean, the fact that an association between *Stylaster* and *Millepora* is presently only known to exist around Iriomote-jima in the Ryukyus Islands offers some support to the theory that island or archipelago environments promote the diversity and abundance of Stylasteridae.

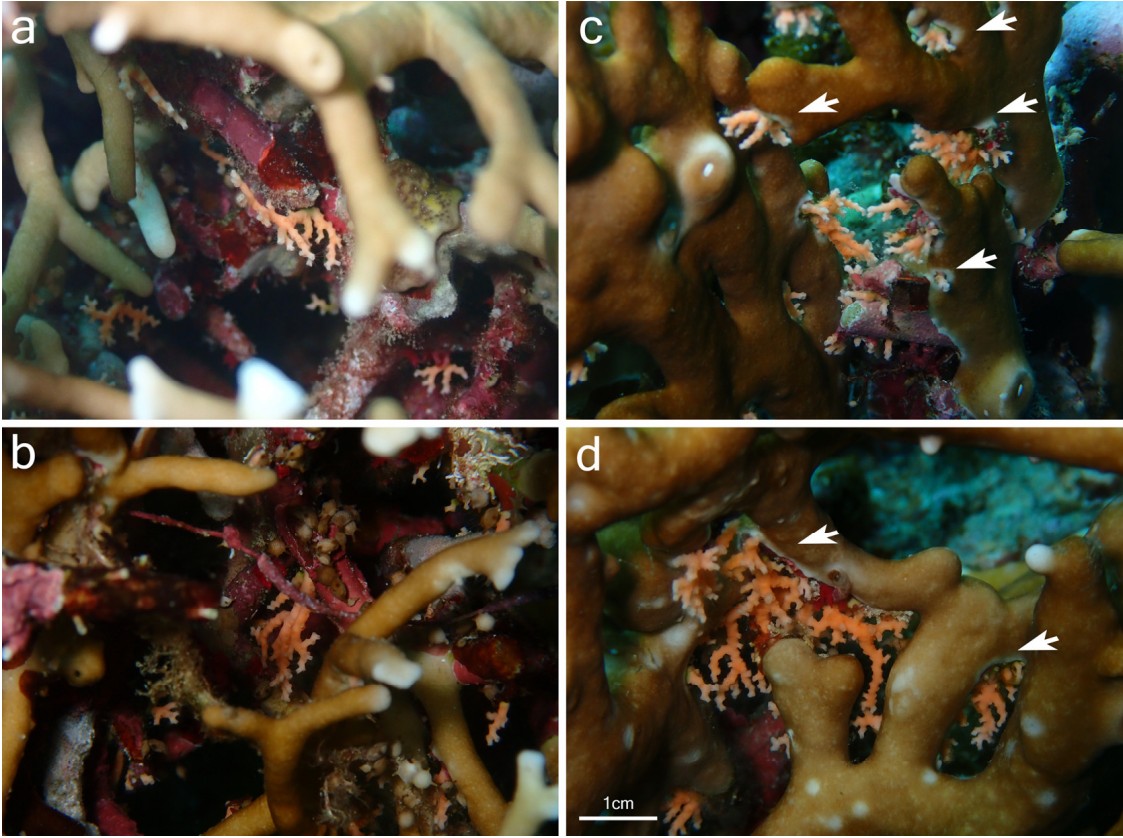

**Figure 4.** *Stylaster* sp. growing (**a**,**b**) on benthic substrate on dead *Millepora* sp. branches; (**c**,**d**) on live *Millepora* branches with a mix of benthic substrate present at the base of *Stylaster* sp. colonies. Bleached *Millepora* tissue surrounding *Stylaster* sp. is indicated by white arrows. Photographs were taken at Nishizaki-Higashi, Iriomote-jima, Japan.

Finally, we also laid transects at four sites to obtain estimates of the prevalence of this association. Although *Millepora* was present at all four sites, where transect data could be obtained, *Stylaster* was only observed at Nakano Beach, where it was hosted by 23.8 ± 21.4% of *Millepora* colonies ($n = 7.3 \pm 6.4$ colonies per transect including 2.6 ± 2.5 colonies inhabited by *Stylaster*, Figure 5). *Millepora* colonies appeared to be more abundant in Nakano Beach (Figure 5). While *Stylaster* inhabiting *Millepora* could potentially be more frequently detected at Nakano Beach due to the abundance of the host; the ANOVA test found no significant differences in the abundance of *Millepora* between sites. Although we also observed the *Millepora* and *Stylaster* relationship in Funauki Port Lighthouse and Nishizaki-Higashi sites, we could not obtain transect data from those sites. The high abundance of *Stylaster* on *Millepora* colonies at Nakano Beach, in contrast with its apparent absence at other nearby

sites, suggests that the association may be constrained to certain environments and that it is patchy in distribution. This is in line with the observations of Montano et al. [6], who recorded the Caribbean association between *Millepora* and *Stylaster* in locations that were very close to other sites where it was not detected. While *Stylaster* has been reported from the Yaeyama Islands [23], their range of habitats in the region are not well documented. Surveys including *Stylaster* potentially living in crevices are required to better understand the context of *Millepora* and *Stylaster* relationship.

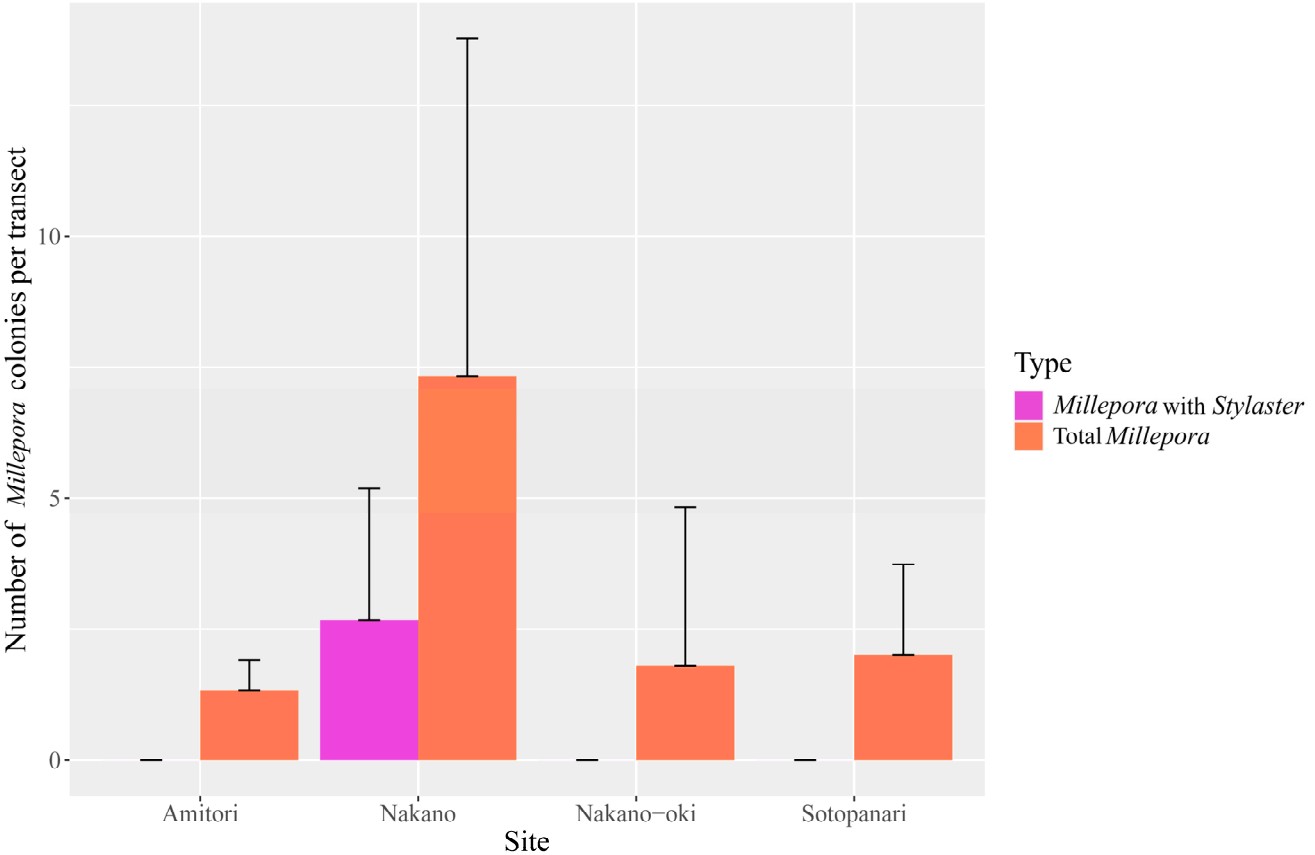

**Figure 5.** Mean abundance of Millepora colonies at each site and Millepora hosting Stylaster at each site, with error bars showing standard deviation.

Our observation of *Stylaster* sp. using *Millepora* spp. as a substrate represents the first record of such an association in the Pacific Ocean. This raises multiple questions regarding the evolutionary implications and the exact meaning of the relationship for both partners. The ecological and physiological conditions in which it can occur, which *Millepora* and *Stylaster* species are involved, and how are they related to their counterparts in the Caribbean, *M. alcicornis*, and *S. roseus*, also remain to be established.

**Author Contributions:** Conceptualization, C.J.L.F. and J.D.R.; Methodology, C.J.L.F. and J.D.R.; Formal Analysis, C.J.L.F., D.P. and J.D.R.; Investigation, C.J.L.F., D.P., G.M.C., E.A.J., I.M. and J.D.R.; Resources, C.J.L.F., G.M.C., E.A.J., I.M. and J.D.R.; Data Curation, C.J.L.F. and J.D.R.; Writing—Original Draft Preparation, C.J.L.F.; Writing—Review and Editing, C.J.L.F., D.P., G.M.C., E.A.J., I.M. and J.D.R.; Visualization, C.J.L.F. and D.P.; Supervision, J.D.R.; Funding Acquisition, J.D.R. All authors have read and agreed to the published version of the manuscript.

**Funding:** C.J.L.F. and E.A.J. are thankful to the 100 Island Challenge Project, Scripps Institution of Oceanography for travel funding and to MEXT for university scholarships. Field work in 2023 was partially supported by a JSPS Grant-in-Aid for Transformative Research Areas entitled "Environmental, ecological, and genetic observations of coral reef Symbiodiniaceae-host holobiont symbioses" (23H03821) to J.D.R.

**Institutional Review Board Statement:** Not applicable.

**Data Availability Statement:** The transect data obtained in this study are available upon request to the corresponding author.

**Acknowledgments:** We thank T. Naruse (U. Ryukyus), N. Holloway, G. Turner and S. Kodera (all Scripps Institute) for field support.

**Conflicts of Interest:** The authors declare no conflicts of interest.

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
