# Peer review of "Millepora spp. as Substrates of Their Hydrozoan Counterparts Stylaster sp. in the Pacific Ocean"

_diversity, doi:10.3390/d16030142_

Round 1

Reviewer 1 Report

Comments and Suggestions for Authors

This is a very interesting study reporting a rare association between two closely related species of hydroids, unique among other hydrozoans in their ability to form calcified exoskeletons. The text is clear, informative and well-written; the aims are clear; the material is well described, the methodology is scientific sound; the results are well presented and the figures are informative; the discussion includes important hypotheses of the nature of the association, that points to possible directions for future study. The report of the diverse forms of association between hydroids and their substrate have been overlooked for a long time in the Hydrozoa literature, and for that I believe this study is of great importance and should be published, as it provides baseline information for further investigations on hydroids diversity and evolution. I have a few minor suggestions for the manuscript, which are provided below.

My main suggestion lies on the possibility of further exploring the data obtained and provided with the paper, as it seems to me that the authors have gathered information that, if maybe further explored, could provide even more insights on this association. The transect data, as mentioned in the material and methods, with information on prevalence of the association, could be explored and included in the form of graphs for visualization of possible patterns of association of theses species in the island. Maybe the prevalence of this association in certain sites of the island could indicate that the occurrence of Stylaster may be better explained by environmental factors than by a specific association between the two species (due to phylogenetically relatedness). Also, the sites with more and less prevalence of the association could be compared in terms of their physical conditions (hydrodynamics, depth, etc.), based on information from the literature, further contributing with this discussion. Additionally, in the discussion, the authors could provide as assessment of the previously known abundance and occurrence of Stylaster in the region, with focus on the different substrates it may have been reported. It would be interesting to know if maybe, in the region, Stylaster also commonly occur on other different substrates (which could indicate the species is a substrate generalist, and its occurrence on Millepora could be better explained by ecology). On the other hand, if Stylaster is only known from the region in association with Millepora, this might indicate a more particular association among these species.

Some minor suggestions for the text:

Line 64: the word “We” in this line should be written in lowercase letter

Line 148: I suggest the term “homologous species” be replaced by the more direct term “closely related species”.

Author Response

We are grateful to Reviewer 1 for the attention given to our manuscript and their valuable suggestions.

My main suggestion lies on the possibility of further exploring the data obtained and provided with the paper, as it seems to me that the authors have gathered information that, if maybe further explored, could provide even more insights on this association. The transect data, as mentioned in the material and methods, with information on prevalence of the association, could be explored and included in the form of graphs for visualization of possible patterns of association of theses species in the island. Maybe the prevalence of this association in certain sites of the island could indicate that the occurrence of Stylaster may be better explained by environmental factors than by a specific association between the two species (due to phylogenetically relatedness). Also, the sites with more and less prevalence of the association could be compared in terms of their physical conditions (hydrodynamics, depth, etc.), based on information from the literature, further contributing with this discussion. Additionally, in the discussion, the authors could provide as assessment of the previously known abundance and occurrence of Stylaster in the region, with focus on the different substrates it may have been reported. It would be interesting to know if maybe, in the region, Stylaster also commonly occur on other different substrates (which could indicate the species is a substrate generalist, and its occurrence on Millepora could be better explained by ecology). On the other hand, if Stylaster is only known from the region in association with Millepora, this might indicate a more particular association among these species.

We thank you for these nice suggestions and encouraging us to further explain and show our data. Following this suggestion, we are adding a new figure in which we have plotted the mean number of Millepora per site, and the mean number of Millepora that hosted Stylaster, and also added a corresponding paragraph in the Discussion. We have also added a sentence about knowledge of shallow water Stylaster in the region, although the information is very scarce. We thank you again for those helpful comments.

Line 64: the word “We” in this line should be written in lowercase letter

We have made the suggested change.

Line 148: I suggest the term “homologous species” be replaced by the more direct term “closely related species”.

We have deleted “homologous” from the text following this suggestion, but refrain from replacing it by “closely related species” as this should be assessed by phylogenetic analyses in the future.

Reviewer 2 Report

Comments and Suggestions for Authors

The manuscript describes observation of stylasterid hydroid Stylaster sp. growing on the colonies of the other hydrozoans coral Millepora. This rare interaction (not investigated) is of interest for ecologists and developmental/cell biologists. But the present manuscript presents only on elementary observation of the phenomenon with some data on occurrence in the new locality. However authors discuss several hypotheses and idea, and these discussions are speculative as they are not based on any real results of investigation.

So I propose to modify the manuscript for publication and a short communication describing the detection of rare association between “hydrozoan corals” in new locality. Nothing more.

Some minor comments to the text.

Line 33 – “ One unusual epibiont of Millepora is...” – just before authors stated: ”... fire corals of the genus Millepora host members of many marine phyla among their associated fauna [2], including cnidarians, crustaceans, molluscs, polychaetes, tunicates, and sponges, among others ” So why Stylaster is called “unusual”? It can be interesting from certain points of view...

Line 64 – “ Furthermore, We also...”

Lines 65-68 – “Although Millepora was present at all four sites where...” – it remains unclear whether Stylaster colonies were present at all sites on natural substrates besides Millepora.

Lines 80-81 – “... if not most Millepora colonies at the three sites where the association was noticed.” – in previous paragraph it was ststed that “...Stylaster was only observed at Nakano Beach, where it was hosted by 23.8±21.4% of Millepora colonies.”

Lines 89-90 – “Therefore, it appears that the colonies of Stylaster sp. can maintain themselves on unhealthy or long dead Millepora colonies.” – What surpised tyhe authors? If Stylaster sp. can be found on natural (abiotic) substrates, the dead Millepora colonies present the same abiotic substrate...

Lines 93-94 – “such as an advantage for planktivory and against intraspecific competition, ...” – the advantage for planktivory and, especially, against intraspecific competition needs explanation.

Line 124 – “...close to Millepora sp. tissue...” – the photo does not allow differentiating between live and dead colonies.

Line 127 – (Fig 3 c, d) – “… base of the Stylaster colonies.” – bleaching of Millepora tissue is evident at the base of Stylaster colonies – it should give evidence for certain antagonistic interaction.

Figures are of good quality.

Author Response

Line 33 – “ One unusual epibiont of Millepora is...” – just before authors stated: ”... fire corals of the genus Millepora host members of many marine phyla among their associated fauna [2], including cnidarians, crustaceans, molluscs, polychaetes, tunicates, and sponges, among others ” So why Stylaster is called “unusual”? It can be interesting from certain points of view...

We have rephrased this statement, replacing “unusual” with “notable”.

Line 64 – “ Furthermore, We also...”

We have changed “We” to lower case.

Lines 65-68 – “Although Millepora was present at all four sites where...” – it remains unclear whether Stylaster colonies were present at all sites on natural substrates besides Millepora.

We thank Reviewer 2 for pointing this out. Unfortunately, Stylaster are often found under overhangs in the coral reefs, and therefore it is difficult to assess their presence based on the transect data that we used. We will consider investigating in the future.

Lines 80-81 – “... if not most Millepora colonies at the three sites where the association was noticed.” – in previous paragraph it was ststed that “...Stylaster was only observed at Nakano Beach, where it was hosted by 23.8±21.4% of Millepora colonies.”

We could record the presence of Stylaster in Nakano Beach in our transects, plus two other sites where we were not able to obtain transect data from, hence the confusion. We have added a sentence to convey information more clearly.

Lines 89-90 – “Therefore, it appears that the colonies of Stylaster sp. can maintain themselves on unhealthy or long dead Millepora colonies.” – What surpised tyhe authors? If Stylaster sp. can be found on natural (abiotic) substrates, the dead Millepora colonies present the same abiotic substrate...

As there is few information available about Stylaster species diversity or habitat range in the Ryukyus, we considered that the status of observed Stylaster sp. as an obligatory or opportunistic symbiont was not established; without further taxonomic investigations we cannot say if the Stylaster we observed is also a species potentially inhabiting crevices of the reefs in the Ryukyus. However, we interpret the presence of Stylaster on healthy, bleached and dead Millepora as an indication that this species is indeed able to live on abiotic and live Millepora substrate. We have clarified our thinking in an additional sentence.

Lines 93-94 – “such as an advantage for planktivory and against intraspecific competition, ...” – the advantage for planktivory and, especially, against intraspecific competition needs explanation.

Thank you for suggesting this. We rephrased the sentence to briefly make the meaning of those hypotheses clearer.

Line 124 – “...close to Millepora sp. tissue...” – the photo does not allow differentiating between live and dead colonies.

Thank you for pointing this out, we have adjusted the size of the image of Fig. 2b (now Fig. 3b) to better show the live tentacles of Stylaster sp.

Line 127 – (Fig 3 c, d) – “… base of the Stylaster colonies.” – bleaching of Millepora tissue is evident at the base of Stylaster colonies – it should give evidence for certain antagonistic interaction.

Thank you for noticing this and pointing this out. We extended our discussion of this a little and edited the figures with arrows to better show the surrounding bleached tissue.

Reviewer 3 Report

Comments and Suggestions for Authors

This manuscript report the relationship between the fire coral Millepora and Stylaster in the Okinawa waters, Japan. The relationship appears to be different from previously reported relationship in the Caribbean. I think this MS is good and can be published after minor revisions. One of the main point can be included in the discussion is, that when the larvae of the Stylaster settle on the Millepora surface, the nematocysts of Millepora must not attack these larvae and even these larvae can tolerate the toxin from Millepora as a condition for the subsequent survival with fire corals. Such example has been demonstrated in the fire coral barnacle Wanella and Millepora relationships. The naupliar larvae of Wanella cannot tolerate the attack by fire corals, so the naupliar of Wanella were killed when approaching the fire corals. However, the cyprids, the final settlement stage, can deactivate the activity of nematocysts and can walk on the surface of fire corals. Even the nematocysts touched the larvae, the larvae did not suffer any damages. Such behavioral control on fire coral hosts may be an essential adaptation for species living with fire corals. I would suggest the author to cite this reference and add in the above points in the discussion. 

Yap F.‐C., J. T. Høeg, B. K. K. Chan (2022). Living on fire: Deactivating fire coral polyps for larval settlement and symbiosis in the fire coral‐associated barnacle. Ecology and Evolution, https://doi.org/10.1002/ece3.9057

The MS has compared the relationship between Stylaster and Millepora in the Pacific and the Carribean Sea. The Stylaster may probably has great diversity and represent cryptic species across the Ocean. Such pattern was seen in the fire coral barnacle Wanella, which composed of a cryptic species complex across the Indo-Pacific region. The authors can consider also citing this pattern and compare with their studied organisms.

https://academic.oup.com/zoolinnean/article/199/4/871/7254671

Author Response

This manuscript report the relationship between the fire coral Millepora and Stylaster in the Okinawa waters, Japan. The relationship appears to be different from previously reported relationship in the Caribbean. I think this MS is good and can be published after minor revisions. One of the main point can be included in the discussion is, that when the larvae of the Stylaster settle on the Millepora surface, the nematocysts of Millepora must not attack these larvae and even these larvae can tolerate the toxin from Millepora as a condition for the subsequent survival with fire corals. Such example has been demonstrated in the fire coral barnacle Wanella and Millepora relationships. The naupliar larvae of Wanella cannot tolerate the attack by fire corals, so the naupliar of Wanella were killed when approaching the fire corals. However, the cyprids, the final settlement stage, can deactivate the activity of nematocysts and can walk on the surface of fire corals. Even the nematocysts touched the larvae, the larvae did not suffer any damages. Such behavioral control on fire coral hosts may be an essential adaptation for species living with fire corals. I would suggest the author to cite this reference and add in the above points in the discussion.

We thank Reviewer 3 for their nice suggestions and the time spent on our manuscript.

Yap F.‐C., J. T. Høeg, B. K. K. Chan (2022). Living on fire: Deactivating fire coral polyps for larval settlement and symbiosis in the fire coral‐associated barnacle. Ecology and Evolution, https://doi.org/10.1002/ece3.9057

Thank you for pointing out this interesting paper to us. This is very relevant to the case of Stylaster and Millepora and we therefore included this reference in our discussion.

The MS has compared the relationship between Stylaster and Millepora in the Pacific and the Carribean Sea. The Stylaster may probably has great diversity and represent cryptic species across the Ocean. Such pattern was seen in the fire coral barnacle Wanella, which composed of a cryptic species complex across the Indo-Pacific region. The authors can consider also citing this pattern and compare with their studied organisms.

https://academic.oup.com/zoolinnean/article/199/4/871/7254671

Thank you also for referring us to this study. This reference shows a pattern of closely related but divergent species in association with Millepora, showing that a number of species in one group take advantage of the symbiotic lifestyle across different regions. We have now incorporated this reference into our manuscript along with a short discussion on the topic.

Round 2

Reviewer 2 Report

Comments and Suggestions for Authors

Lines 94-98: The sentence "This reinforces the idea that Stylaster sp. mainly uses Millepora as a substrate, that the species involved..." looks unmatched. Maybe it could be better: " This reinforces the idea that while Stylaster sp. mainly uses Millepora as a substrate, that the species involved..."

Lines 198-220: - long discussion having no relation to the topic. It is possible simply to mention that there was no possibility to identify species, but some morphological features of the colony allows to suppose that the species are different from Caribbean ones.  

Lines 233-251: this part remains too speculative, I propose to remove it completely.